# Blood-stage immunity to *Plasmodium chabaudi* malaria following chemoprophylaxis and sporozoite immunization

Wiebke Nahrendorf[1,2]*[†‡], Philip J Spence[1†‡], Irene Tumwine[1§], Prisca Lévy[1§], William Jarra[1], Robert W Sauerwein[2], Jean Langhorne[1]*[§]

[1]Division of Parasitology, MRC National Institute for Medical Research, London, United Kingdom; [2]Department of Medical Microbiology, Radboud University Medical Centre, Nijmegen, Netherlands

*For correspondence: Wiebke. Nahrendorf@ed.ac.uk (WN); jlangho@nimr.mrc.ac.uk (JL)

Present address: [†]Institute of Immunology and Infection Research, University of Edinburgh, Edinburgh, United Kingdom; [‡]Centre for Immunity, Infection and Evolution, University of Edinburgh, Edinburgh, United Kingdom; [§]Mill Hill Laboratories, Francis Crick Institute, London, United Kingdom

**Competing interests:** The authors declare that no competing interests exist.

**Abstract** Protection against malaria in humans can be achieved by repeated exposure to infected mosquito bites during prophylactic chloroquine treatment (chemoprophylaxis and sporozoites (CPS)). We established a new mouse model of CPS immunization to investigate the stage and strain-specificity of malaria immunity. Immunization with *Plasmodium chabaudi* by mosquito bite under chloroquine cover does not generate pre-erythrocytic immunity, which is acquired only after immunization with high sporozoite doses. Instead, CPS immunization by bite elicits long-lived protection against blood-stage parasites. Blood-stage immunity is effective against a virulent, genetically distinct strain of *P. chabaudi*. Importantly, if exposure to blood-stage parasitemia is extended, blood-stage parasites induce cross-stage immunity targeting pre-erythrocytic stages. We therefore show that CPS immunization can induce robust, long-lived heterologous blood-stage immunity, in addition to protection against pre-erythrocytic parasites following high dose sporozoite immunization. Cross-stage immunity elicited by blood-stage parasites may further enhance efficacy of this immunization regimen.

## Introduction

Protective immunity against microorganisms is developed after repeated infection and recovery (*Mutapi et al., 2013*). Vaccines are usually successful if they mimic these naturally acquired immune responses (*Fox, 1984*; *Mielcarek et al., 2006*). While protection against viruses and bacteria can be induced by vaccination with killed (inactivated) or live-attenuated pathogens (*Delany et al., 2014*), there is no licensed vaccine for human parasitic diseases like malaria that pose a major global health burden.

The apicomplexan malaria parasite *Plasmodium* is transmitted by bites of female anopheline mosquitoes. In the vertebrate host, sporozoites injected into the dermis migrate to the liver, where they establish a clinically silent infection of hepatocytes. Merozoites are then released from the liver and invade erythrocytes, leading to an exponential asexual replication cycle that is entirely responsible for the clinical signs and symptoms associated with malaria. Immunity against severe disease can be acquired following repeated infection, but sterile parasite clearance is rarely achieved (*Goncalves et al., 2014*). Clinically immune adults in endemic areas still harbor parasites in their blood-stream (*Okell et al., 2009*). These asymptomatic carriers also develop gametocytes, the form transmissible to mosquitoes, thereby allowing the parasite to complete its life cycle.

To achieve malaria control and eventually eradication, transmission must be blocked (*Kappe et al., 2010*). A vaccine that protects against pre-erythrocytic parasites, and thus outperforms naturally

**eLife digest** Malaria is a life-threatening infectious disease in humans that is caused by a single-celled parasite called *Plasmodium*. The parasite is carried between people by mosquitos; when an infected mosquito bites a human, the parasite is injected into the bloodstream with the mosquito's saliva. *Plasmodium* first infects liver cells but then re-enters the bloodstream, where it infects red blood cells leading to symptoms of disease. If another mosquito bites the infected individual at this so-called 'blood-stage', the parasite can be passed to this mosquito and the cycle of transmission continues.

Currently there are no vaccines available that can effectively protect against malaria. Although an experimental vaccine containing a weakened form of the parasite can protect against the liver-stage parasites, it fails to prevent the parasite from multiplying in the red blood cells. Therefore, the individuals remain susceptible to severe malaria.

Recently, researchers have developed a new strategy for immunization that provides exposure to both liver-stage and blood-stage parasites. Human volunteers taking an anti-malarial drug were deliberately exposed to mosquitos carrying the parasite on three separate occasions. Although the volunteers were infected with the parasite, the anti-malarial drug killed the parasites inside the red blood cells. After the end of the drug treatment, the volunteers were exposed to mosquitos carrying the parasite and they were still protected from infection. These results are promising, but it is not clear if the volunteers have acquired immunity to liver-stage or blood-stage parasites, or even both.

To answer this important question, Nahrendorf et al. developed a similar immunization strategy in mice. Just like the human volunteers, the mice were treated with an anti-malarial drug and exposed to mosquitos carrying *Plasmodium* on three separate occasions. Although the immunizations did not protect the mice against early infection in the liver, they did provide long-term protection against parasites multiplying in the red-blood cells.

The immunity generated by this immunization strategy also protected the mice against another strain of *Plasmodium*, different to the one used in the immunizations. The experiments also show that prolonged exposure to the blood-stage parasites can even lead to immunity against the liver-stage parasites.

Nahrendorf et al.'s findings show that this immunization strategy can protect individuals against both the liver-stage and blood-stage parasites. The next challenges are to find out how the immunity generated by one stage of infection can protect against the other stages, and to discover which molecules on the parasite the immune system targets.

acquired immunity, would greatly facilitate this aim. Parasites that arrest during the liver stage, either because of irradiation (*Nussenzweig et al., 1967*) or targeted gene deletion (*Mueller et al., 2005*), can provide immunity against challenge infection. Indeed, immunization of human volunteers with irradiated sporozoites can induce sterile protection in experimental settings (*Clyde et al., 1973*; *Seder et al., 2013*). However, in the absence of acquired immunity to the blood-stage parasite a pre-erythrocytic vaccine that is only partially effective, and therefore permits breakthrough erythrocytic infections, will provide no protection against severe malaria (*Bejon et al., 2011*). The inclusion of a blood-stage component together with an effective pre-erythrocytic vaccine is therefore preferred to provide a multi-stage malaria vaccine that minimizes both transmission and disease (*Ellis et al., 2010*; *Goodman and Draper, 2010*).

A recently described experimental malaria immunization protocol using chemoprophylaxis and sporozoites (CPS) (*Roestenberg et al., 2009*) ensures exposure to pre-erythrocytic and blood-stage parasites, and hence has the unique potential to induce protection against all *Plasmodium* life cycle stages in the vertebrate host. Three immunizations with bites of 10–15 *Plasmodium falciparum*-infected mosquitoes under chloroquine chemoprophylaxis are sufficient to elicit sterile protection against homologous challenge in human volunteers (*Roestenberg et al., 2009*; *Bijker et al., 2013*, *2014*). Although it is not possible to measure liver parasite burden in human volunteers directly, it appears that immunity exclusively targets pre-erythrocytic parasite life cycle stages, as there is no protection against direct blood challenge (*Bijker et al., 2013*). CPS immunization is thus substantially more effective than immunization with irradiated sporozoites, which requires 1000 mosquito bites

(*Clyde et al., 1973*) or five intravenous (*iv*) injections of more than 100,000 sporozoites for sterile protection (*Seder et al., 2013*). We therefore hypothesize that transient blood-stage parasitemia, before abrogation by chloroquine, may contribute to immunity following CPS immunization.

In this study, we have investigated the stage- and strain-specificity of protection in a novel mouse model of CPS immunization using *Plasmodium chabaudi*. *P. chabaudi* establishes a chronic, non-lethal blood-stage infection, which has been used extensively to characterize the immune response to blood-stage parasites in vivo (*Stephens et al., 2012*). A recently optimized protocol for *P. chabaudi* mosquito transmission (*Spence et al., 2012*) allows us now to also study pre-erythrocytic stages of this rodent parasite. Heterologous protection can readily be assessed since many genetically distinct *P. chabaudi* isolates displaying a variety of virulence phenotypes are available (*Mackinnon and Read, 1999*; *Otto et al., 2014*). We immunized C57BL/6 mice three times with bites of *P. chabaudi*-infected mosquitoes under oral chloroquine chemoprophylaxis similar to human clinical trials (*Roestenberg et al., 2009*; *Bijker et al., 2013*, *2014*). This approach is unique amongst all published animal models of CPS immunization (*Beaudoin et al., 1977*; *Golenser et al., 1977*; *Orjih et al., 1982*; *Belnoue et al., 2004*; *Friesen and Matuschewski, 2011*; *Inoue et al., 2012*; *Nganou-Makamdop et al., 2012b*; *Doll et al., 2014*; *Lewis et al., 2014*; *Peng et al., 2014*), which have (without exception) used *iv* injection of high numbers of *Plasmodium berghei* or *Plasmodium yoelii* sporozoites for immunization. Furthermore, rather than evaluating effector mechanisms by challenging shortly after immunization (*Beaudoin et al., 1977*; *Belnoue et al., 2004*; *Friesen and Matuschewski, 2011*), we performed the challenge 100 days after the final immunization to test the generation and maintenance of long-term immunological memory.

CPS immunization with *P. chabaudi* by mosquito bite does not generate pre-erythrocytic immunity, which is acquired only after immunization with high doses of sporozoites. Instead, immunization by bite elicits blood-stage immunity that is effective against the immunizing strain and also a more virulent, genetically distinct *P. chabaudi*. Moreover, extended exposure to blood-stage parasitemia elicits robust pre-erythrocytic immunity, comparable to protection afforded by high dose sporozoite immunization. Exposure to blood-stage parasites thus elicits heterologous blood-stage immunity and can contribute to the pre-erythrocytic efficacy of this immunization regimen. Therefore, these findings add significantly to advances from previous CPS immunization mouse models by evaluating the generation of immune memory after immunization with *P. chabaudi* by mosquito bite. This is relevant for our understanding of acquired immunity in a malaria endemic setting, and can inform multi-stage malaria vaccine development.

## Results

We investigated the stage- and strain-specificity of protection in a novel mouse model of CPS immunization (*Figure 1*). C57BL/6 mice were immunized three times at 2-week intervals with *P. chabaudi* AS-infected mosquito bites. For certain experimental questions, it was necessary to deviate from the natural route of infection and inject sporozoites *iv* to control the dose. Starting on the day of infection, mice were then treated orally with 100 mg per kg chloroquine for 10 days following each immunization. To assess the long-term efficacy of acquired immunity, mice were challenged approximately 100 days after the last immunization. Protection against pre-erythrocytic or blood-stage parasites was evaluated after mosquito bite or direct blood challenge.

### CPS immunization leads to a transient blood-stage infection

Transient blood-stage parasitemia is a key feature of CPS immunization. We used quantitative RealTime (qRT) PCR to measure erythrocytic parasite burden after each immunization with *P. chabaudi* AS-infected mosquito bites under chloroquine cover. After the first immunization, approximately 50,000 parasites per ml whole blood were detected within the first erythrocytic cycle (*Figure 2*). The amount of blood-stage parasites within the first erythrocytic cycle varied extensively (median 47,596, range 67–222,699), reflecting the stochastic inoculation of sporozoites during mosquito bite (*Beier et al., 1991*; *Ponnudurai et al., 1991*; *Medica and Sinnis, 2005*). Thereafter, chloroquine reduced parasitemia by 86–96% every 24 hr. After the fourth cycle, the majority of erythrocytic parasites were cleared. Similarly, after the second and third immunization mice experienced a substantial number of circulating blood-stage parasites for 48–72 hr. However, although blood-stage parasites were detected in all but one mouse, parasitemia in the first

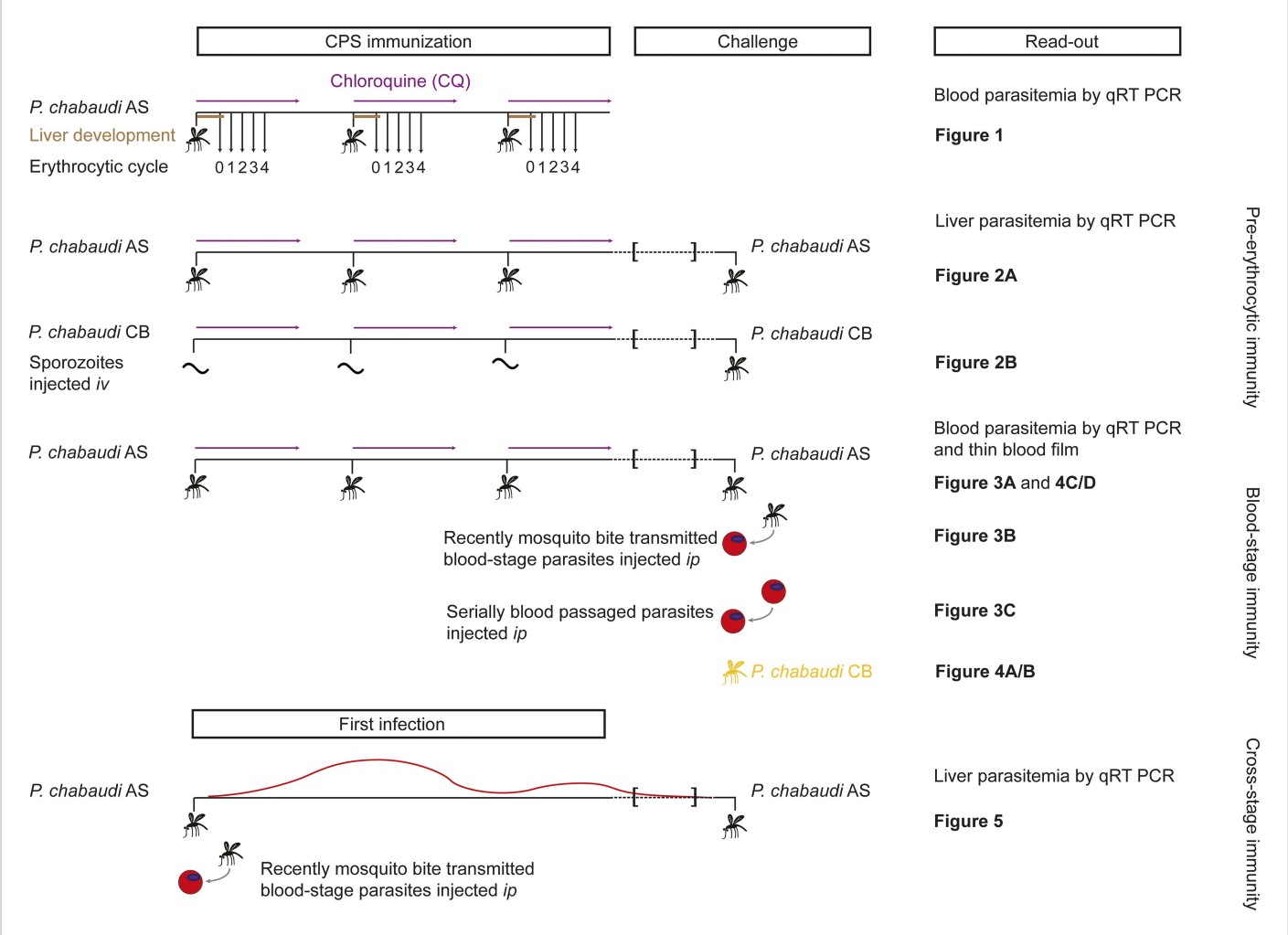

**Figure 1**. Overview of experimental procedures. To quantify transient blood-stage exposure during chemoprophylaxis and sporozoite (CPS) immunization female C57BL/6 were immunized three times at 2-week intervals with *P. chabaudi* AS-infected mosquito bites (typically 9.15 [median, range 6.9–13.6] [*Spence et al., 2012*]). Following each immunization mice received 100 mg per kg chloroquine (CQ) *per os* daily for 10 days, starting from the day of infection. A small blood sample was taken 48 hr after each mosquito transmission (before merozoite egress from the liver, *P. chabaudi* develops in the liver for 52 hr [*Stephens et al., 2012*]; erythrocytic cycle 0), and then every 24 hr until erythrocytic replication cycle 4. Blood parasitemia was analyzed by sensitive quantitative RealTime (qRT) PCR (*Figure 2*). Pre-erythrocytic immunity was evaluated in mice immunized three times with either *P. chabaudi* AS-infected mosquito bites (*Figure 3A*) or by intravenous (*iv*) injection of defined numbers of *P. chabaudi* CB sporozoites (*Figure 3B*). *P. chabaudi* CB was used since mosquitoes infected with this parasite harbor an increased number of sporozoites in their salivary glands (*Spence et al., 2012*), which made injections of high numbers of sporozoites technically feasible. Mice were challenged 100 days after the last immunization by mosquitoes infected with the respective homologous strain. Liver parasitemia was examined 42 hr after challenge by qRT PCR. Blood-stage immunity was assessed in mice immunized with *P. chabaudi* AS-infected mosquito bites by qRT PCR and thin blood film following homologous challenge with either infected mosquito bites (*Figures 4A, 5C/D*) or intraperitoneal (*ip*) injection of parasitized erythrocytes, which were either derived from a donor mouse infected by mosquito bite (recently mosquito transmitted, *Figure 4B*) or after 26–32 serial blood-passages (*Figure 4C*). Heterologous protection was assessed using *P. chabaudi* CB-infected mosquito bites (*Figure 5A/B*). To evaluate cross-stage protection mice received a first infection with *P. chabaudi* AS either by mosquito bite or by *ip* injection of recently mosquito transmitted parasitized erythrocytes. The resulting blood-stage infection was eventually self-cured without intervention. Mice were re-challenged 100 days after their first infection with *P. chabaudi* AS-infected mosquito bites and liver parasitemia was evaluated by qRT PCR (*Figure 6*).

erythrocytic cycle was reduced by 5- and 13-fold after the second and third immunizations, respectively, when compared to infection controls (*Figure 2*). Consequently, one CPS immunization is sufficient to reduce blood-stage parasite burden within the first erythrocytic cycle, which indicates either pre-erythrocytic or blood-stage immunity.

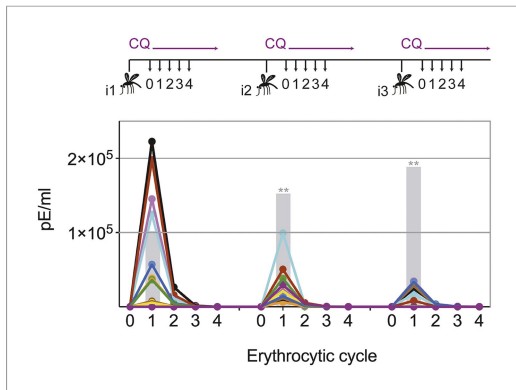

**Figure 2**. Chloroquine permits transient blood-stage parasitemia during each immunization. The number of parasitized erythrocytes (pE) per ml of whole blood was enumerated by quantitative RealTime PCR after each CPS immunization (i1, i2, i3) with *P. chabaudi* AS-infected mosquito bites under chloroquine (CQ) cover. The number of pE (at the late trophozoite stage) was quantified immediately before merozoite egress from the liver, at 48 hr post mosquito transmission (erythrocytic cycle 0), and then every 24 hr until erythrocytic replication cycle 4. Daily parasitemia of 10 CPS immunized mice (each color represents an individual mouse) are shown. Blood-stage parasites were detected within the first erythrocytic cycle after every immunization in all but one mouse after the final immunization. Gray bars represent the mean parasitemia in the first erythrocytic cycle of naive mice infected as controls for mosquito transmission efficiency separate with each immunization (n = 3–5). Significant differences in the number of blood-stage parasites in the first erythrocytic cycle between naive and CPS immunized mice are indicated (Mann Whitney test, **p ≤ 0.01).

## Pre-erythrocytic immunity requires high doses of sporozoites during CPS immunization

In order to assess directly whether pre-erythrocytic immunity was generated by this CPS immunization protocol, liver parasite burden was analyzed after mosquito bite challenge. Surprisingly, there was no difference in the liver parasite burden between mice given three CPS immunizations with *P. chabaudi* AS-infected mosquito bites and infection controls (*Figure 3A*). Therefore, immunization by mosquito bite under chloroquine cover according to this protocol failed to elicit pre-erythrocytic immunity. This is in contrast to results from other animal models of CPS immunization where pre-erythrocytic immunity is induced after high numbers of sporozoites are injected *iv* (*Belnoue et al., 2004*; *Friesen and Matuschewski, 2011*; *Inoue et al., 2012*; *Nganou-Makamdop et al., 2012b*; *Lewis et al., 2014*; *Peng et al., 2014*). To test if pre-erythrocytic immunity can be induced after CPS immunization with large numbers of sporozoites, we used *P. chabaudi* CB, since mosquitoes infected with this parasite strain harbor an increased number of sporozoites in their salivary glands compared to *P. chabaudi* AS (*Spence et al., 2012*), making these experiments technically feasible. In agreement with previous studies (*Belnoue et al., 2004*; *Friesen and Matuschewski, 2011*; *Inoue et al., 2012*; *Nganou-Makamdop et al., 2012b*; *Lewis et al., 2014*; *Peng et al., 2014*), mice immunized *iv* three times with 10,000 *P. chabaudi* CB sporozoites under chloroquine cover did show reduced liver parasite burden (up to 90%) after mosquito bite challenge, compared to infection controls (*Figure 3B*). Conversely, mice immunized *iv* three times with a low dose of 100 *P. chabaudi* CB sporozoites (representative of the estimated number of *P. chabaudi* sporozoites that initiate infection via mosquito bite, *Spence et al., 2012*) do not acquire pre-erythrocytic immunity (*Figure 3B*). It also appears that CPS immunization with 10,000 live sporozoites was more effective at inducing pre-erythrocytic immunity than immunization with 10,000 irradiated *P. chabaudi* CB sporozoites, which arrest during hepatic development (*Suhrbier et al., 1990*) and do not establish a blood-stage infection (*Figure 3B*). This suggests that complete liver-stage maturation and the increased blood-stage parasitemia that accompanies immunization with 10,000 sporozoites (as compared to 100 sporozoites or mosquito bite) could contribute to pre-erythrocytic protection.

## CPS immunization by mosquito bite elicits blood-stage immunity

To test whether the transient blood-stage infection resulting from CPS immunization by mosquito bite (*Figure 2*) is sufficient to induce protection against erythrocytic parasites, we challenged mice approximately 100 days after the last immunization and measured blood-stage parasitemia (*Figure 4*). Mice that were CPS immunized with *P. chabaudi* AS had similar numbers of parasitized erythrocytes as compared to mock immunized controls within the first five erythrocytic cycles following mosquito bite challenge (*Figure 4A*). However, from erythrocytic cycle 6 parasitemia was significantly reduced and blood-stage parasites were cleared more rapidly in immunized mice. This was reflected in a 6-fold

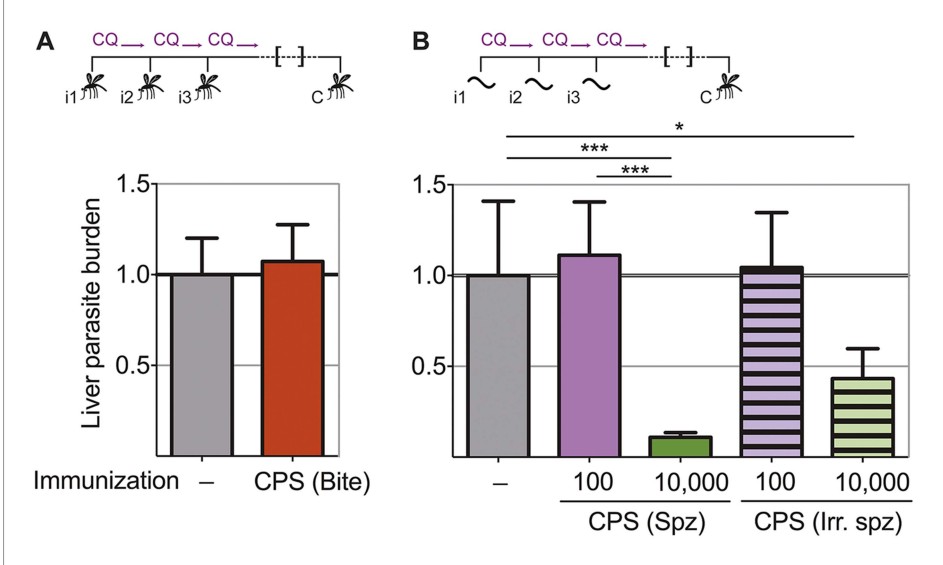

**Figure 3**. Pre-erythrocytic immunity following CPS immunization requires high doses of sporozoites. Liver parasite burden was determined 42 hr after mosquito bite challenge as copy number of *P. chabaudi*-specific 18S rRNA. (**A**) Mice were CPS immunized three times (i1, i2, i3) with *P. chabaudi* AS-infected mosquito bites under chloroquine (CQ) cover (CPS (Bite)) and challenged (C) 96–104 days after immunization by bites of *P. chabaudi* AS-infected mosquitoes (pooled data from three independent experiments; naive infection controls (−) n = 25, CPS (Bite) n = 35). (**B**) 100 or 10,000 untreated or irradiated (Irr.) *P. chabaudi* CB sporozoites (spz) were injected *iv* three times under CQ cover. Mice were challenged 96 days after immunization by bites of *P. chabaudi* CB-infected mosquitoes (naive infection controls (−) n = 12, all other groups n = 20). All data are displayed relative to the mean of corresponding liver parasite burden of naïve infection controls and presented as mean ± SEM, (**A**) Mann–Whitney test: no significant difference between the groups; (**B**) Kruskal Wallis with Dunn's multiple comparisons test *p ≤ 0.05, ***p ≤ 0.001.

reduction of total area under the curve (AUC) (right inset, *Figure 4A*). Protection against erythrocytic parasites was also evaluated by direct blood challenge, using blood-stage parasites obtained from a donor mouse infected by mosquito bite. Similar to the results of mosquito bite challenge, blood-stage parasitemia was significantly reduced (*Figure 4B*), but the infection was still chronic in some mice (*Figure 4—figure supplement 1*). However, blood-stage protection was abrogated when CPS immunized mice were challenged with serially blood passaged (SBP) *P. chabaudi*; blood-stage parasites with increased virulence following multiple passages through naive mice (*Spence et al., 2013*, *Figure 4C*). In this case, CPS immunization reduced blood-stage parasite burden only between erythrocytic cycle 6 and 8, as compared to mock immunized controls. This was reflected in only a 1.2-fold reduction in total AUC (right inset, *Figure 4C*). Therefore, CPS immunization by mosquito bite elicits homologous blood-stage immunity, which is most effective in the context of mosquito transmission.

## CPS immunization elicits heterologous blood-stage immunity

CPS immunization has so far not been shown to induce protection against challenge with genetically distinct strains of *Plasmodium*; a situation that would be encountered in human malaria-endemic areas. The genetic diversity amongst strains of *P. chabaudi* (*Mackinnon and Read, 1999*; *Otto et al., 2014*) allows us to investigate heterologous immunity in this model of CPS immunization. Mice that were CPS immunized with *P. chabaudi* AS-infected mosquito bites had reduced peak parasitemia, and blood-stage parasites were cleared faster, when compared to mock immunized mice after homologous (*P. chabaudi* AS) and heterologous (*P. chabaudi* CB) mosquito bite challenge (*Figure 5A–D*). A direct comparison between *P. chabaudi* AS and CB challenge revealed nonetheless that homologous blood-stage immunity is more effective than heterologous immunity. In this experiment, CPS immunization reduced total AUC by 25-fold following homologous challenge, as

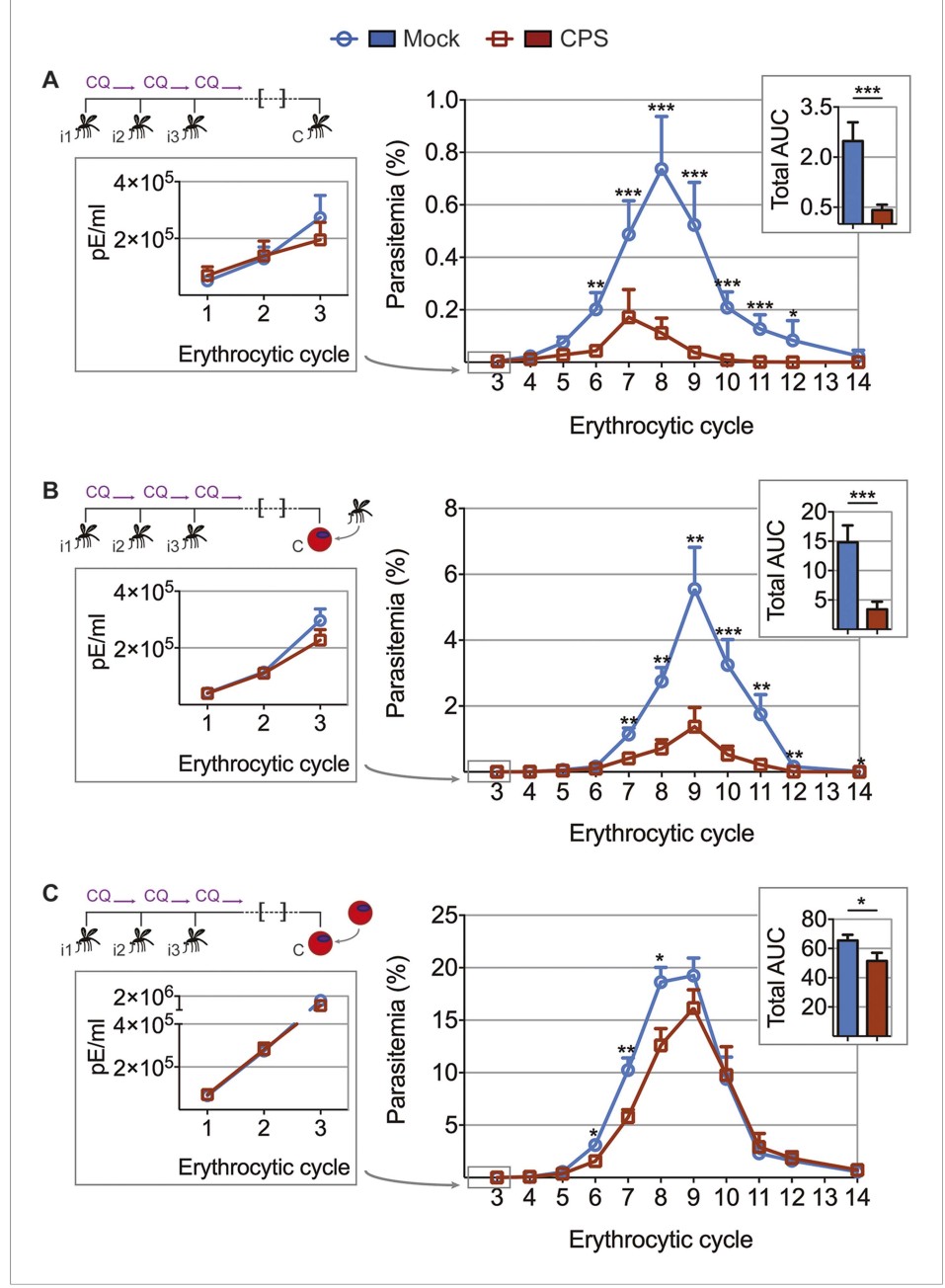

**Figure 4**. CPS immunization elicits blood-stage immunity. Mice were CPS immunized three times (i1, i2, i3) using chloroquine (CQ) and *P. chabaudi* AS-infected or uninfected mosquito bites (mock immunized). Approximately 100 days after the final CPS immunization, mice were challenged (C) with *P. chabaudi* AS. Erythrocytic parasitemia was evaluated daily by quantitative RealTime PCR (cycle 1–3, displayed as parasitized erythrocytes (pE) per ml whole blood; left) and from cycle 3–14 by thin blood-film (expressed as % parasitized erythrocytes [parasitemia] 0.01% parasitemia corresponds to 1,000,000 pE per ml; middle). The total area under the curve (AUC) was calculated for each mouse between erythrocytic cycle 3 and 14 (right). (**A**) Mosquito bite challenge: parasitemia from 1st to 3rd (n = 10) and between 3rd and 14th erythrocytic cycle (representative of three independent experiments, n = 12–19), total AUC between cycle 3 and 14 (n = 19). (**B**) Direct blood challenge using 10,000 erythrocytic parasites obtained from a donor mouse infected by mosquito bite; injected intraperitoneal (*ip*): parasitemia between 1st and 3rd (n = 10) and 3rd and 14th erythrocytic cycle (representative of three independent experiments, n = 10), total AUC between cycle 3 and 14 (n = 10). (**C**) Blood challenge using 10,000 serially blood passaged parasites; injected *ip*: parasitemia from 1st to 3rd (n = 10) and between 3rd and 14th erythrocytic cycle (representative of two independent experiments,
*Figure 4. continued on next page*

*Figure 4. Continued*

n = 8–10), total AUC between cycle 3 and 14 (n = 10). All data are presented as mean ± SEM, Mann–Whitney test per time point *p ≤ 0.05, **p ≤ 0.01, ***p ≤ 0.001.

The following figure supplement is available for figure 4:

**Figure supplement 1**. Chronic blood-stage infection in CPS immunized mice.

---

compared to 6-fold following heterologous challenge (*Figure 5E*). Nevertheless, CPS immunization elicits blood-stage protection against a robust heterologous challenge with the genetically distinct, and more virulent, CB strain of *P. chabaudi*.

## Extended exposure to blood-stage parasites elicits robust pre-erythrocytic immunity

Blood-stage parasites appear to be both the source and target of protection following CPS immunization with *P. chabaudi* AS-infected mosquito bites. It was shown that immunization with serially blood passaged *P. yoelii* parasites with prophylactic chloroquine treatment can elicit pre-erythrocytic immunity (*Belnoue et al., 2008*). We wanted to assess whether blood-stage parasites have the potential to induce pre-erythrocytic protection also in the context of mosquito transmission. We therefore asked whether a fulminant blood-stage infection could elicit cross-stage immunity against pre-erythrocytic parasites. We infected mice with *P. chabaudi* AS by mosquito bite or intraperitoneal (*ip*) injection of recently mosquito transmitted blood-stage parasites. The two groups of mice were not drug-treated, and therefore experienced a low-grade chronic, recrudescing blood-stage infection for up to 90 days (*Spence et al., 2013*). After mosquito bite challenge, both groups of mice demonstrated reduced liver parasite burden (up to 85%), compared to infection controls (*Figure 6*). Cross-stage immunity is therefore a powerful mechanism for protection against pre-erythrocytic parasites, which may be absent during CPS immunization with small sporozoite numbers as the blood-stage infection is curtailed by the use of chloroquine.

## Discussion

Timing, route of infection, and antigen dose play major roles in determining the initial priming of the antimalarial immune response (*Legorreta-Herrera et al., 2004*; *Elliott et al., 2005*; *Belnoue et al., 2008*; *Guilbride et al., 2012*; *Nganou-Makamdop et al., 2012a*). We incorporated many of these aspects from human clinical trials (*Roestenberg et al., 2009*; *Bijker et al., 2013*, *2014*) in a new *P. chabaudi* mouse model of CPS immunization to investigate the stage- and strain-specificity of CPS-induced protection against malaria. Our results highlight the complexity of immunity against the different life cycle stages of the malaria parasite (*Table 1*). Pre-erythrocytic immunity appears to depend on the number of immunizing sporozoites. In this study, we find no evidence of pre-erythrocytic immunity after CPS immunization with *P. chabaudi* infected mosquito bites, which inoculate an estimated maximum of 100 sporozoites per immunization (*Spence et al., 2012*, *2013*). Sterile pre-erythrocytic protection was however reported in human CPS immunization trials (*Roestenberg et al., 2009*; *Bijker et al., 2013*, *2014*). *Anopheles stephensi* mosquitoes experimentally infected with *P. falciparum* 3D7 or NF54 habor 50–200 times more sporozoites in their salivary glands (*Bijker et al., 2013*) compared to *P. chabaudi* AS (*Spence et al., 2012*). However, only few sporozoites are injected into the skin during mosquito bite and this number is independent of salivary gland sporozoites load (*Beier et al., 1991*; *Ponnudurai et al., 1991*; *Medica and Sinnis, 2005*). Since the number of sporozoites establishing a liver-stage infection can further be influenced by inherent sporozoite infectivity (*Khan and Vanderberg, 1991*), our best estimate of immunizing sporozoite dose is the number of infected erythrocytes observed directly after egress from the liver. Assuming that 10,000 merozoites are released from one infected liver cell (*White et al., 2014*), we estimate approximately 400 infected hepatocytes (95% CI 137–1250) in human volunteers (*Bijker et al., 2013*) compared to 5 in *P. chabaudi* AS immunized mice (95% CI 1–31). Therefore, the number of infected hepatocytes after CPS immunization by mosquito bite in the *P. chabaudi* mouse model is approximately 100-fold lower than in CPS immunized humans. We show that this differences in

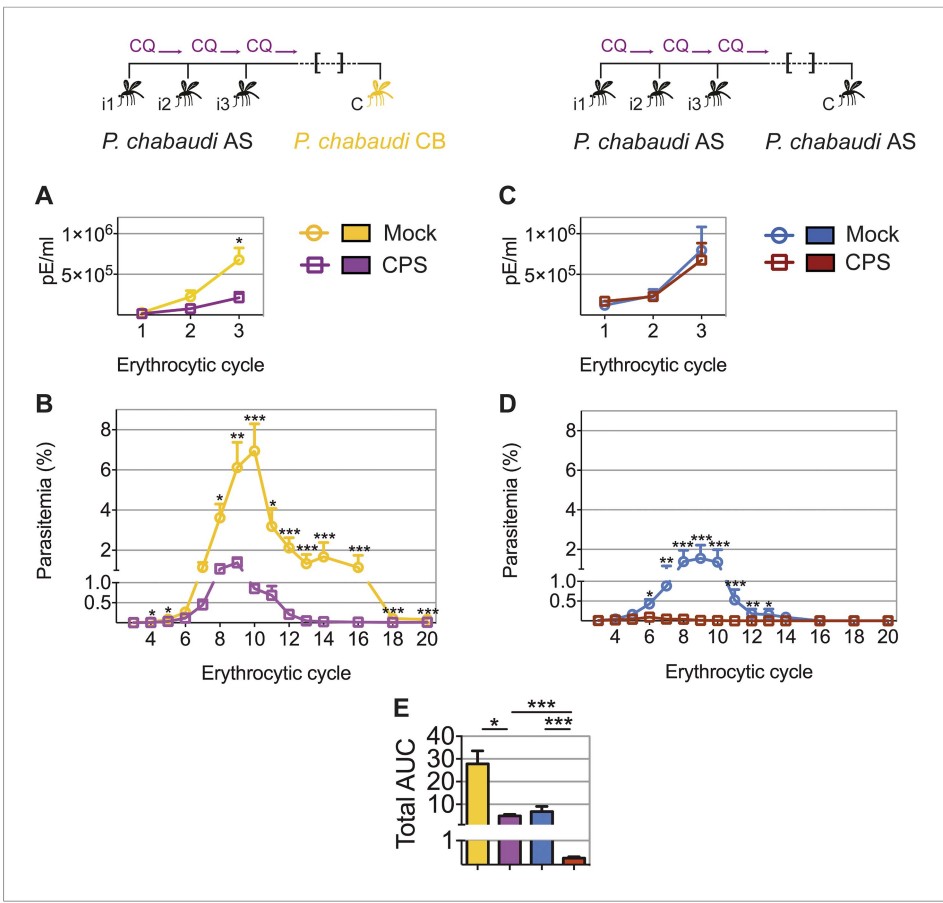

**Figure 5**. CPS immunization elicits heterologous blood-stage immunity. Mice were CPS immunized three times (i1, i2, i3) under chloroquine (CQ) cover by *P. chabaudi* AS-infected mosquito bites or mock immunized with uninfected mosquito bites, and challenged (C) 96–107 days later by mosquito bite. Erythrocytic parasitemia was evaluated daily by quantitative RealTime PCR for blood-stage parasites (cycle 1–3, displayed as parasitized erythrocytes (pE) per ml whole blood) and from cycle 3–20 by thin blood-film (expressed as % parasitized erythrocytes [parasitemia], 0.01% parasitemia corresponds to 1,000,000 pE per ml). (**A/B**) Heterologous challenge with *P. chabaudi* CB infected mosquitoes (**A**) Parasitemia between 1st and 3rd (n = 10) and (**B**) from 3rd to 20th erythrocytic cycle (n = 20). (**C/D**) Homologous challenge with *P. chabaudi* AS-infected mosquitoes (**C**) Parasitemia between 1st and 3rd (n = 10) and (**D**) from 3rd to 20th erythrocytic cycle (CPS immunized n = 20, mock immunized n = 19). (**E**) Total AUC comparing mock and CPS immunized mice receiving heterologous or homologous mosquito bite challenge. Data are presented as mean ± SEM, (**A–D**) Mann–Whitney test per time point (**E**) Kruskal Wallis with Dunn's multiple comparisons test *$p \leq 0.05$, **$p \leq 0.01$, ***$p \leq 0.001$.

infected hepatocyte numbers by a factor of 100 can be significant for the development of pre-erythrocytic immunity: three immunizations with 10,000 *P. chabaudi* sporozoites *iv* induce long-lasting protection against mosquito bite challenge, while three immunizations with 100 *P. chabaudi* sporozoites fail to do so. This is in general agreement with other rodent malaria studies using *P. berghei* (*Beaudoin et al., 1977*; *Golenser et al., 1977*; *Orjih et al., 1982*; *Friesen and Matuschewski, 2011*; *Nganou-Makamdop et al., 2012b*; *Lewis et al., 2014*) or *P. yoelii* (*Belnoue et al., 2004*; *Inoue et al., 2012*; *Doll et al., 2014*; *Peng et al., 2014*), in which sterile pre-erythrocytic immunity is observed after immunization with high sporozoite doses (typically 10,000–50,000 sporozoites per immunization), while a reduction in sporozoite dose or the number of immunizations leads to breakthrough blood-stage infections upon challenge (*Belnoue et al., 2004*; *Inoue et al., 2012*). Reduction of the number of *P. falciparum*-infected mosquitoes also reduces the frequency of sterilely protected volunteers (*Bijker et al., 2014*), indicating that the number of sporozoites establishing a liver-stage infection fails to surpass the protective threshold to elicit sterile

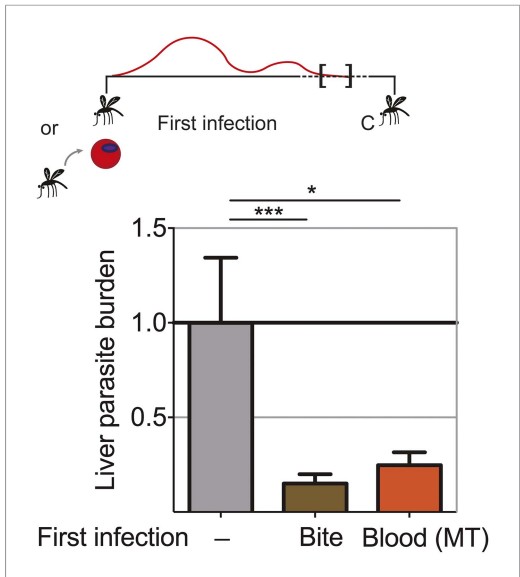

**Figure 6**. Extended blood-stage parasitemia elicits preerythrocytic immunity. Mice received a first infection with *P. chabaudi* AS either by mosquito bite (Bite) or intraperitoneal (*ip*) injection of 10,000 parasitized erythrocytes obtained from a donor mouse infected by mosquito bite (Blood (MT)). Blood-stage parasitemia was self-cured before challenge (C; 98 days postinfection) using *P. chabaudi* AS-infected mosquito bites. Liver parasite burden was determined 42 hr after mosquito bite challenge as copy number of *P. chabaudi*-specific 18S rRNA. Data are displayed relative to the mean of corresponding liver parasite burden of naive infection controls (–). Pooled data from two independent experiments (Naive (–) and Bite n = 30, Blood (MT) n = 20) are presented as mean ± SEM, Kruskal Wallis with Dunn's multiple comparisons test *p ≤ 0.05, ***p ≤ 0.001.

pre-erythrocytic immunity. This may also explain why in malaria-endemic areas pre-erythrocytic immunity is thought to be absent (*Tran et al., 2013*). In addition to maximizing specific responses against immunodominant antigens, CPS immunization with high numbers of sporozoites may broaden the immune repertoire by including protective responses against subdominant antigens, which could enhance heterologous pre-erythrocytic protection (*Trieu et al., 2011*). This may further be enhanced by the longer liver-stage development of *P. falciparum* (egress 6.8 days after mosquito bite, *Roestenberg et al., 2012*) compared to *P. chabaudi* (egress after 52 hr, *Stephens et al., 2012*). Longer liver stage development may positively influence the generation of pre-erythrocytic immunity by allowing time for protective immune responses to develop.

A key feature of CPS immunization is that it permits exposure to all *Plasmodium* life cycle stages in the vertebrate host, including parasitized erythrocytes. It is well known that repeated exposure to blood-stage parasites, for example, in rodent models (*Jarra and Brown, 1985*; *Legorreta-Herrera et al., 2004*; *Elliott et al., 2005*), in humans exposed to ultra-low doses of parasitized erythrocytes while receiving drug treatment (*Pombo et al., 2002*), during malariatherapy of neurosyphilis patients (*Collins and Jeffery, 1999*), and in people living in malariaendemic areas (*Bull et al., 1998*), induces bloodstage immunity. Our results also clearly show that during CPS immunization repeated transient blood-stage infection (less than 0.01% parasitemia for 48–72 hr) elicits long-lasting bloodstage immunity. Protection against challenge infection was only apparent after multiple erythrocytic replication cycles and patent blood-stage parasite densities, which could indicate that bloodstage protection was not observed in CPS immunized human volunteers after direct blood-challenge because drug treatment is required at low parasite densities as soon as patency is reached (typically between the third and fourth erythrocytic cycle, *Bijker et al., 2013*). Blood-stage parasites are however recognized in human volunteers, which was demonstrated by an earlier increase of IFNγ and monokines induced by IFNγ (MIG) concentrations in CPS immunized volunteers compared to infection controls after direct blood challenge (*Bijker et al., 2013*). It will be of value to investigate whether the observed blood-stage protection in this mouse model is also detected in CPS immunized primates, where a longer blood-stage infection than that allowed in human clinical trials is possible.

Despite the reduced peak parasitemia and faster clearance of blood-stage parasites during the acute phase of infection in CPS immunized mice, recrudescent parasitemia could still be observed in the chronic phase of infection after challenge, suggesting a parasite variant escapes the protective immune response (*McLean et al., 1982*). There are very few reports on protective efficacy of CPS immunization (or indeed any sporozoite-based vaccine) against direct blood-challenge (*Belnoue et al., 2004*; *Inoue et al., 2012*; *Peng et al., 2014*). *Doll et al. (2014)* reported that sustained, subpatent blood-stage infection after treatment with a commonly used dose of chloroquine can induce partial blood-stage protection. Low-grade transient blood-stage parasitemia, achieved by attenuation of blood-stage parasites using an antimalarial drug (*Pombo et al., 2002*;

**Table 1**. Relationship between the dose of immunizing pre-erythrocytic and blood-stage parasites and the acquisition of immunity.

|  | Low dose | High dose |
| --- | --- | --- |
| Sporozoites (liver-stage parasites) | Mosquito bite or 100 sporozoites *iv*: no pre-erythrocytic immunity (**Figure 3A/B**) | 10,000 sporozoites *iv*: pre-erythrocytic immunity (**Figure 3B**) |
| Blood-stage parasites | Blood-stage infection curtailed by chloroquine: partial blood-stage immunity (**Figure 4**) | Fulminant, self-cured blood-stage infection: blood-stage immunity (**Spence et al., 2013**); pre-erythrocytic immunity (**Figure 6**) |

*Elliott et al., 2005*), a DNA alkylating agent (*Good et al., 2013*) or genetic tools (*Ting et al., 2008*; *Aly et al., 2010*) similarly provides protection against homologous and heterologous blood-stage challenge.

We could show that CPS-induced blood-stage immunity is effective against heterologous mosquito bite challenge with a more virulent and genetically distinct strain of *P. chabaudi* (*Mackinnon and Read, 1999*; *Lamb and Langhorne, 2008*). In agreement with the observed cross-species protection after CPS immunization with mefloquine (*Inoue et al., 2012*), and one study using chemically attenuated sporozoites for immunization (*Purcell et al., 2008*), heterologous protection is less effective than homologous immunity. Nevertheless, CPS immunization can elicit long-lived protection against both homologous and heterologous blood-stage parasites, which will be important to minimize disease severity in the case of breakthrough blood-stage infections. This is essential for the development of an effective multi-stage malaria vaccine (*Ellis et al., 2010*; *Bejon et al., 2011*).

In stark contrast to the observed heterologous blood-stage protection after mosquito bite challenge infection, protection was almost completely abrogated following direct blood-challenge with virulent parasites obtained after continuous serial blood passage. This suggests that blood-stage parasites immediately after mosquito transmission express antigens not present on serially blood passaged parasites and that these antigens may be the target of protective immunity following CPS immunization. Serially blood passaged parasites can hence escape from CPS-induced blood-stage immunity. One group of *Plasmodium* genes, whose expression is altered by mosquito transmission during blood-stage infection is the *Plasmodium* interspersed repeat gene family (*pir*); termed *cir* in *P. chabaudi* (*Lawton et al., 2012*). Transcription of more than half of all *cir* genes is increased in blood-stage parasites after mosquito transmission compared with their transcription after serial blood passage. This diversification of *cir* transcription is associated with a more effective host immune response, which in turn attenuates parasite virulence (*Spence et al., 2013*). The *cir* genes could also be candidate targets for cross-stage immunity, as *P. berghei pir* genes are also transcribed during the liver stages (personal communication BM Franke-Fayard and CJ Janse, Leiden University Medical Center, The Netherlands). An investigation into shared PIR proteins between liver and blood-stage parasites may hence provide valuable information for multi-stage malaria vaccine development. Furthermore, the absence of blood-stage protection in previous CPS models may have been due to challenge with serially blood passaged rather then recently mosquito transmitted blood-stage parasites. It is therefore always essential to evaluate blood-stage immunity in the context of mosquito transmission.

As shared antigenic targets between liver- and blood-stage parasites have been described (*Tarun et al., 2008*), the exciting possibility of cross-stage protection has been considered but only rarely assessed. Genetically modified *fabb/f-* sporozoites that arrest late in liver-stage development protect against *iv* challenge with 100 blood-stage parasites (*Butler et al., 2011*). On the other hand, a blood-stage infection with serially blood passaged *P. yoelii*, drug-treated with chloroquine after 4–5 days, reduces liver parasite load upon *iv* challenge with 35,000 sporozoites (*Belnoue et al., 2008*). In our model of CPS immunization by mosquito bite, it is likely that repeated transient blood-stage infection during immunization elicits the observed blood-stage protection, although we cannot yet exclude that responses acquired against pre-erythrocytic antigens, which are shared with blood-stage parasites (*Tarun et al., 2008*), contribute as well. Because of the low number of infected hepatocytes after CPS immunization with *P. chabaudi*-infected mosquito bites this seems however unlikely. Nevertheless, we demonstrate unequivocally that extended exposure to blood-stage parasites is an effective

stimulator of pre-erythrocytic immunity. Exposure to blood-stage parasites during CPS immunization may thus significantly contribute to the observed pre-erythrocytic protection in human volunteers (*Roestenberg et al., 2009*; *Bijker et al., 2013*, *2014*). Indeed, cross-stage immunity could be responsible for the unprecedented efficacy of CPS immunization compared to immunization with irradiated sporozoites (*Clyde et al., 1973*; *Seder et al., 2013*), which arrest early during liver-stage development and never establish a blood-stage infection. While extending exposure to replicating blood-stage parasites by delayed drug administration is not possible in humans, the incorporation of chemically (*Good et al., 2013*) or genetically (*Ting et al., 2008*; *Aly et al., 2010*) attenuated blood-stage parasites should be considered to further enhance the generation and maintenance of both pre-erythrocytic and blood-stage immunity. This makes CPS immunization a powerful tool for the development of an effective multi-stage malaria vaccine.

## Materials and methods

### Mice

Inbred C57BL/6 mice, originally obtained from Jackson Laboratories (Bar Harbor, USA), were bred under specific pathogen-free conditions at the MRC National Institute for Medical Research (NIMR) for over 30 years. All experiments were performed in accordance with UK Home Office regulations (PPL 80/2358) and approved by the ethical review panel at the MRC NIMR. Mice were housed under reverse light conditions (light 19.00–07.00, dark 07.00–19.00 GMT) at 20–22°C and 50% relative humidity, with continuous access to mouse breeder diet and water.

### Parasites and mosquitoes

*Plasmodium chabaudi chabaudi* (*P. chabaudi*) AS and CB were cloned at the University of Edinburgh and sent to the NIMR in 1978 and 1982, respectively. Both parasite lines were routinely serially blood-passaged (SBP) through mice between 26 and 32 times by *ip* injection of parasitized erythrocytes or mosquito transmitted according to a recently published protocol (*Spence et al., 2012*). In brief C57BL/6 mice were injected *ip* with 100,000 parasitized erythrocytes and 14 days post infection gametocytemia was assessed on Giemsa-stained (VWR, Lutterworth, UK) thin blood film. *A. stephensi* mosquitoes, pre-treated with 50 µg/ml gentamicin (Sigma, Gillingham, UK) and starved for 24 hr before transmission, were fed on anaesthetised mice with >0.1% gametocytes of total erythrocytes at a ratio of >1 mouse per 100 mosquitoes. Mosquitoes were kept at 26.0°C (± 0.5°C) in an ultrasonic humidity cabinet and provided with 8% Fructose and 0.05% 4-Aminobenzoic acid (both Sigma, Gillingham, UK) feeding solution. After 8 days, a sample of 20 mosquitoes were dissected to assess development of *P. chabaudi* oocyts in the midgut. For infection of experimental mice 20–23 mosquitoes were transferred into 25 cl paper cups after 14 days, starved for 24 hr and fed on anaesthetized mice for 20–25 min at room temperature. Typically, mice were exposed to 9.15 (median, range 6.9–13.6) *P. chabaudi*-infected mosquito bites (*Spence et al., 2012*).

### Isolation of sporozoites

Sporozoites were isolated from *P. chabaudi*-infected mosquito salivary glands 15 or 16 days post gametocyte feed. Salivary glands were dissected under a stereomicroscope, transferred to a glass homogenizer and kept in RPMI supplemented with 0.2% Glucose, 0.2% Sodium bicarbonate (both Sigma, Gillingham, UK), 2 mM L-Glutamine (Gibco, Paisley, UK) and 10% fetal bovine serum (GE Healthcare Life Sciences, Pittsburgh, Pennsylvania), on ice for maximum 2 hr. Sporozoites were released from the glands by gentle homogenization and washed three times before enumeration. The number of sporozoites per infected mosquito was enumerated for each mosquito transmission experiment. For *iv* injection *P. chabaudi* CB sporozoites were used, since mosquitoes infected with this parasite strain harbor an increased number of sporozoites per infected mosquito in their salivary glands (median 1638, range 175–2576) compared to *P. chabaudi* AS (median 438, range 43–956) (*Spence et al., 2012*).

### Attenuation of sporozoites by irradiation

To arrest parasite development in the early stages of liver development *P. chabaudi* CB infected *A. stephensi* were exposed to 16 Gray (=16,000 rad) (*Nganou-Makamdop et al., 2012b*) of Caesium-137 γ-irradiation 15 or 16 days post gametocyte feed just prior to sporozoite dissection.

## Chemoprophylaxis and sporozoite (CPS) immunization and challenge regimen

Female age-matched 8–10 week old C57BL/6 mice were infected three times in 2-week intervals with *P. chabaudi*: either by *P. chabaudi* AS-infected mosquito bites or *iv* injection of *P. chabaudi* CB sporozoites. Mice were treated after each immunization with 100 mg/kg chloroquine diphosphate salt (chloroquine, Sigma, Gillingham, UK) by gavage daily for 10 days, starting from the day of mosquito transmission. Mock immunized mice received uninfected mosquito bites and chloroquine treatment. 100 days after the last CPS immunization mice were challenged with *P. chabaudi* AS or *P. chabaudi* CB-infected mosquito bites, or via *ip* injection of 100,000 parasitized erythrocytes (direct blood-challenge) that were obtained from either a donor mouse infected by mosquito bite or after serial blood passage. Since each erythrocytic cycle of *P. chabaudi* is approximately 24 hr long (*Stephens et al., 2012*) development of blood-stage parasitemia was monitored daily by microscopy of Giemsa-stained thin blood films, from erythrocytic cycle 3 to 14 and every other day thereafter. The limit of detection was 0.01% parasitemia, which equals 1 parasitized red blood cell in 10,000 erythrocytes or 1,000,000 parasitized erythrocytes per ml of blood.

## Quantification of liver- and blood-stage parasitemia by quantitative Real Time PCR

Liver and blood-parasitemia was assessed by quantifying 18S rRNA using qRT PCR. 42 hr after mosquito bite challenge mice were terminally anaesthetized and immediately upon cessation of respiration their livers were perfused with 5 ml RNAse-free Phosphate buffered saline (PBS, Gibco, Paisley, UK). Using the Gentle MACS homogenizer (Miltenyi, Bisley, UK) the whole liver was homogenized in 4 ml Guanidinium thiocyanate (Sigma, Gillingham, UK) solution (*Chomczynski and Sacchi, 2006*), and 600 µl aliquots were stored at −80°C. To assess blood-parasite burden during CPS immunization and in the first erythrocytic cycles following bite challenge 10 µl of blood were isolated from the tip of the mouse-tail and after two washes in RNAse-free PBS stored at −80°C in 100 µl Guanidinium thiocyanate solution (*Chomczynski and Sacchi, 2006*). The first sample was taken either just before (erythrocytic cycle 0) or 20 hr after liver merozoite egress (erythrocytic cycle 1) and then every 24 hr for 4 days. Since *P. chabaudi* displays a synchronous infection (*Stephens et al., 2012*) all blood-stage parasites analyzed were therefore at the late trophozoite stage of development. RNA was extracted from liver- as well as blood-samples using the Guanidinium-thiocyanate-phenol-chlorophorm method (all Sigma, Gillingham, UK; *Chomczynski and Sacchi, 2006*). RNA was thereafter reverse transcribed by PCR (temperature profile: 25°C for 10 min, 42°C for 20 min, 98°C for 5 min) using 75U MuLV Reverse Transcriptase, 30U RNAse Inhibitor, and 2.5 µM Random Hexamer primers (all Applied Biosystems, Paisley, UK) per sample. The amount of 18S rRNA copies was quantified by Real-Time PCR using TaqMan Universal PCR Master Mix (Applied Biosystems, Paisley, UK), 300 ηM forward primer (5′-AAGCATTAAATAAAGCGAATACATCCTTAT-3′), 300 ηM reverse primer (5′-GGGAGT TTGGTTTTGACGTTTATGCG-3′), and 50 ηM probe ([6FAM]CAATTGGTTTACCTTTTGCTCTTT[TAM]). All reactions were performed in the ABI 7900 HT Real Time PCR machine (temperature profile: 50°C for 2 min, 95°C for 10 min, 40 cycles of 95°C for 15 s and 60°C for 1 min). The amount of parasite 18S rRNA in the liver was calculated based on a Standard curve of known copy numbers of 18S rRNA. For every experiment liver-parasite burden was normalized to the mean burden of controls infected in the same experiment. Blood-stage parasitemia was quantified based on a Standard curve of 10-fold dilutions of mosquito transmitted *P. chabaudi* AS late trophozoites prepared identically to the samples.

### Statistical analysis

Data were analyzed using GraphPad Prism v7. Unpaired data between two groups at a specific time point were analyzed by Mann–Whitney test (two-tailed, non-parametric). Differences between more than two groups were analyzed by non-parametric Kruskal–Wallis test with Dunn's multiple comparisons test. Significant differences are indicated by asterisks with *$p \leq 0.05$, **$p \leq 0.01$, ***$p \leq 0.001$.

### Acknowledgements

We thank Sarah Keller McLaughlin for skilled technical assistance, and we are grateful for support from the Division of Biological Services, MRC NIMR. Anja Scholzen and Else M Bijker are acknowledged for

critical reading of the manuscript. WN, PJS, IT, PL, WJ, and JL were supported by the UK Medical Research Council (U117584248). This study was also funded by the FP7 founded European Virtual Institute of Malaria Research (EVIMalaR, grant agreement number 242095) and the Wellcome Trust (UK). WN received an EVIMalaR PhD scholarship. PJS was the recipient of a Leverhulme Trust Early Career Fellowship.

## Additional information

### Funding

| Funder | Grant reference | Author |
| --- | --- | --- |
| Leverhulme Trust | | Philip J Spence |
| Medical Research Council (MRC) | U117584248 | Wiebke Nahrendorf, Philip J Spence, Irene Tumwine, Prisca Lévy, William Jarra, Jean Langhorne |
| Wellcome Trust | | Philip J Spence, Prisca Lévy, Jean Langhorne |

The funders had no role in study design, data collection and interpretation, or the decision to submit the work for publication.

### Author contributions

WN, PJS, Conception and design, Acquisition of data, Analysis and interpretation of data, Drafting or revising the article; IT, PL, WJ, Acquisition of data, Analysis and interpretation of data, Drafting or revising the article; RWS, JL, Conception and design, Analysis and interpretation of data, Drafting or revising the article

### Ethics

Animal experimentation: All experiments were performed in accordance with UK Home Office regulations (PPL 80/2358) and approved by the ethical review panel at the MRC National Institute for Medical Research.

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
