## [Decision Letter]

Thank you for sending your work entitled “Blood-stage immunity to *Plasmodium chabaudi* malaria following chemoprophylaxis and sporozoite immunization” for consideration at *eLife*. Your article has been favorably evaluated by Prabhat Jha (Senior editor) and 4 reviewers, one of whom is serving as a guest Reviewing editor.

The following individuals responsible for the peer review of your submission have agreed to reveal their identity: Urszula Krzych (Reviewing editor) and Michael Good (peer reviewer). Two other reviewers remain anonymous.

The Reviewing editor and the other reviewers discussed their comments before we reached this decision, and the Reviewing editor has assembled the following comments to help you prepare a revised submission.

The study reported here was considered interesting and reviewers pointed to the novel aspect of your study in that using CPS in the *P. chabaudi* represents a somewhat new murine model. The improvement to mosquito transmission of *P. chabaudi* offers new insights that are distinct from those seen with the *P. berghei* and *P. yoelii* models. On the whole, the study was well performed, and for the most part, the paper is well written. What follows is a condensed version of suggestions and substantive concerns expressed by the reviewers:

1) The discordant results showing pre-erythrocytic stage protection in the CPS murine model system and CPS studies conducted in humans warrant a thorough discussion regarding these differences. As an example, immunizations of human subjects via mosquito bite with *P. falciparum* via under CPS coverage induces sterile pre-erythrocytic immunity (7), which has not been observed in the mice without *intravenous* (*iv*) administration of very high doses of sporozoites. Is this related to the *P. falciparum* versus *P. chabaudi* sporozoite inoculum delivered by mosquito bite? Although the answer to the question as to why there is a difference between rodent and humans remains unanswered, omitting this discussion diminishes the importance of the study.

2) The aim of a blood-stage vaccine is to ultimately protect malaria-naïve individuals from clinical disease. The observation that that blood-stage exposure provided protection against pre-erythrocytic parasites (Figure 6) raised several concerns:

a) Although noteworthy, extending blood-stage exposure to enhance CPS immunization efficacy does not appear to be a transferable strategy to humans (in the final part of the Discussion section). With a delayed drug administration to humans (as has been achieved here in mice, as refereed to in the Discussion) one would risk that humans would be displaying malaria symptoms. These issues need to be addressed and the limitations acknowledged.

b) Protection against pre-erythrocytic parasites afforded by blood stage infection without chemoprophylactic drug treatment (Figure 6) argues against the relevance of the CPS immunization model which does not provide pre-erythrocytic immunity after immunization by mosquito bite and therefore questions the ultimate message of the manuscript. The authors may consider rewriting these aspects and broaden the Discussion.

c) While these cross-stage protection experiments are of great interest and significance to the malaria field, the results shown in Figure 6 have been considered as “stand alone” results. Perhaps this aspect of the study could be extended and additional experiments testing for immunologic mechanisms included. This addition would improve the quality of the manuscript.

3) CPS treatment gives rise to heterologous blood stage immunity. It is not clear from this study whether this is due to cross-stage immunity, or due to the low grade blood parasitemia that follows CPS and which would be in keeping with previous studies showing that exposure to very low density blood infections can induce heterologous blood stage immunity. While blood stage immunity has not been shown in human studies with *P. falciparum* and CPS, it quite plausible that it does exist for *P. falciparum* as well but it is simply undetectable because of the need to treat volunteers soon after the appearance of blood stage parasites. Please expand this point in your Discussion.

4) The CPS immunization protocol in mice protects against blood-stage infection (including heterologous strain) only when the challenge is delivered by mosquito bite (and not serial blood passage). This suggests immunity is raised against variant antigens that are shared between the heterologous strains following mosquito bite delivery. Although the data show protection against these genetically distinct strains of *P. chabaudi*, the potential for immune escape suggested by these results warrants further discussion.

5) A careful editing of the manuscript was suggested to avoid a possible confusion:

a) In the Introduction, the authors state that CPS provides both heterologous and pre-erythrocytic immunity but later state that CPS “abrogate” pre-erythrocytic immunity (in the end of the Results section).

b) A disunity in the message is that low dose CPS immunization by mosquito bite offers no pre-erythrocytic protection but some homologous or heterologous erythrocytic immunity; high dose sporozoite challenge *iv*. or no drug, “self-cured” mosquito bite or *ip* blood stage infection provide pre-erythrocytic immunity.

---

## [Author Response]

*The study reported here was considered interesting and reviewers pointed to the novel aspect of your study in that using CPS in the* P. chabaudi *represents a somewhat new murine model. The improvement to mosquito transmission of* P. chabaudi *offers new insights that are distinct from those seen with the* P. berghei *and* P. yoelii *models. On the whole, the study was well performed, and for the most part, the paper is well written. What follows is a condensed version of suggestions and substantive concerns expressed by the reviewers*:

*1) The discordant results showing pre-erythrocytic stage protection in the CPS murine model system and CPS studies conducted in humans warrant a thorough discussion regarding these differences. As an example, immunizations of human subjects via mosquito bite with* P. falciparum *via under CPS coverage induces sterile pre-erythrocytic immunity (Bijker, 2013), which has not been observed in the mice without* iv *administration of very high doses of sporozoites. Is this related to the* P. falciparum *versus* P. chabaudi *sporozoite inoculum delivered by mosquito bite? Although the answer to the question as to why there is a difference between rodent and humans remains unanswered, omitting this discussion diminishes the importance of the study*.

We carried out CPS immunization of C57BL/6 mice using *P. chabaudi* AS-infected *Anopheles stephensi* mosquito bites, similar to human volunteers exposed to *P. falciparum* 3D7 or NF54 infected *A. stephensi* bites (49, 7, 8). This means that the exact number of sporozoites inoculated, and subsequently the number of infected hepatocytes remains unknown. *P. chabaudi* AS-infected mosquitoes harbor a median number of 438 (range 43-956) sporozoites in their salivary glands (52). These numbers are in the same order of magnitude as *P. falciparum* infection rates observed in wild mosquitoes ([Bibr bib47a], [3], [Bibr bib51a]), but 50 to 200 times lower than infection rates of experimentally infected *A*. *stephensi* used for human CPS immunizations (7).

However, injection of sporozoites during mosquito bite is a stochastic process and does not necessarily correlate with the salivary gland load of the mosquito (3, 47, 35). Typically few sporozoites (estimated between 1 and 100) are injected during one infectious bite (3). Based on the number of infected erythrocytes observed directly after release from the liver and assuming that 10,000 merozoites are released from a single infected liver cell (60), we estimate that approximately 5 hepatocytes (95% confidence interval (CI) 1-31) are infected with *P. chabaudi* AS, whereas in human CPS approximately 400 hepatocytes are infected after the first immunization (95% CI 137-1250, [7]). This may also be influenced by the higher number of infectious mosquitoes (12 to 15, [49], [7], [8]) in humans compared to a typical *P. chabaudi* mosquito transmission (Median 9.15 bites, [52]) and possibly low infectivity of *P. chabaudi* sporozoites in C57BL/6 mice, as described previously for *P. berghei* in BALB/c mice (28). Therefore, we estimate that the number of infected hepatocytes after CPS immunization with mosquito bites in the *P. chabaudi* mouse model is approximately 100-fold lower than in human CPS.

We show that this differences in infected hepatocyte numbers by a factor of 100 can be significant for the development of pre-erythrocytic immunity: Three immunizations with 10,000 *P. chabaudi* sporozoites *iv* induces long lasting protection against mosquito bite challenge, while three immunizations with 100 *P. chabaudi* sporozoites fails to do so (Figure 3).

The greater number of infected hepatocytes may broaden protective pre-erythrocytic immune responses by including subdominant antigens (59). This may further be enhanced by the longer liver-stage development of *P. falciparum* in human CPS (egress 6.8 days after mosquito bite, [50]) compared to rodent malaria species (*P. chabaudi* egress after 52h, [54]). A longer liver stage development may positively influence the generation of pre-erythrocytic immunity by allowing time for protective immune responses to develop.

Thus the lower number of infected hepatocytes after *P. chabaudi* infection by mosquito bite and the shorter liver-stage development may result in a less efficient induction of protective immunity compared to CPS immunization with *P. falciparum* in humans.

These considerations have been added to the Discussion.

*2) The aim of a blood-stage vaccine is to ultimately protect malaria-naïve individuals from clinical disease. The observation that that blood-stage exposure provided protection against pre-erythrocytic parasites (*Figure 6*) raised several concerns*:

*a) Although noteworthy, extending blood-stage exposure to enhance CPS immunization efficacy does not appear to be a transferable strategy to humans (in the final part of the Discussion section). With a delayed drug administration to humans (as has been achieved here in mice, as refereed to in the Discussion) one would risk that humans would be displaying malaria symptoms. These issues need to be addressed and the limitations acknowledged*.

That blood-stage parasites can convey pre-erythrocytic immunity is an exciting possibility for the induction of protective immunity to malaria and should be explored further, including in non-human primates and ultimately in humans (as mentioned in the Discussion). Of course we did not mean to suggest extending exposure to replicating blood-stage parasites in human volunteers beyond patency on thick blood film, which would be associated with major clinical risks. Instead it may be worth considering genetically (57, 1) or chemically-attenuated (22) blood-stage parasites, which would ensure prolonged exposure to infected erythrocytes, while posing minimal risks for the volunteer.

This ambiguity has now been rectified in the Abstract and in the Introduction. The Discussion was extended to incorporate these points.

*b) Protection against pre-erythrocytic parasites afforded by blood stage infection without chemoprophylactic drug treatment (*Figure 6*) argues against the relevance of the CPS immunization model which does not provide pre-erythrocytic immunity after immunization by mosquito bite and therefore questions the ultimate message of the manuscript. The authors may consider rewriting these aspects and broaden the Discussion*.

While CPS immunization using *P. chabaudi* AS infected mosquito bites does not induce protection against liver-stage parasites (Figure 3), it was shown that sterile pre-erythrocytic protection can be acquired after CPS immunization of human volunteers with *P. falciparum* (7). As discussed in comment #1 this is likely explained by longer liver-stage development and a 100-fold higher number of infected hepatocytes, which translate into exponentially more blood-stage parasites being released from the liver in human clinical trials. Here we demonstrate that exposure to a large amount of blood-stage parasites gives rise to robust pre-erythrocytic immunity (Figure 6). This suggests that pre-erythrocytic protection after CPS immunization of human volunteers may, at least in part, be mediated by exposure to blood-stage parasites during immunization under chloroquine cover. Cross-stage immunity could be responsible for the unprecedented efficacy of CPS immunization compared to immunization with irradiated sporozoites (12, 51), which arrest early during liver-stage development and never establish a blood-stage infection.

Moreover using our novel *P. chabaudi* mouse model of CPS immunization we show that in addition to pre-erythrocytic protection after immunization with large numbers of sporozoites (Figure 3) partial blood-stage immunity can be acquired (Figure 4), which cannot be investigated in human volunteers. Therefore, given exposure to sufficient numbers of sporozoites and blood-stage parasites, CPS immunization can elicit powerful pre-erythrocytic as well as blood-stage immunity (Table 1). CPS immunization is therefore an invaluable tool to explore the generation of antimalarial immunity against multiple parasite life cycle stages, which can empower multi-stage malaria vaccine development.

The Discussion has been broadened to highlight these arguments.

*c) While these cross-stage protection experiments are of great interest and significance to the malaria field, the results shown in*
Figure 6
*have been considered as “stand alone” results. Perhaps this aspect of the study could be extended and additional experiments testing for immunologic mechanisms included. This addition would improve the quality of the manuscript*.

As pointed out in comment #2b our finding that exposure to blood-stage parasites can induce pre-erythrocytic immunity underpins the hypothesis that pre-erythrocytic immunity observed in humans after CPS immunization may be partly generated by exposure to blood-stage parasites during immunization. These results are therefore central for the scope of this paper.

We are pleased that the reviewers also believe, as we do, that “cross-stage protection experiments are of great interest and significance for the malaria field”. A detailed investigation of immunological mechanisms is of course necessary; however, an in depth study of cross-stage immunity using a panel of knock-out mice and/or deletion of specific subsets of cells would be beyond the scope of this publication. Since we use mosquito transmission and rest the mice after primary infection for 100 days, one experiment to evaluate immunological memory responses will last for 5 months. To uncover the immunological mechanism comprehensively would therefore take at least one, more likely two years. The authors hope the reviewers and editors understand the need to leave this exciting study into the mechanisms of cross-stage immunity for future investigation.

*3) CPS treatment gives rise to heterologous blood stage immunity. It is not clear from this study whether this is due to cross-stage immunity, or due to the low grade blood parasitemia that follows CPS and which would be in keeping with previous studies showing that exposure to very low density blood infections can induce heterologous blood stage immunity. While blood stage immunity has not been shown in human studies with *P. falciparum *and CPS, it quite plausible that it does exist for *P. falciparum *as well but it is simply undetectable because of the need to treat volunteers soon after the appearance of blood stage parasites. Please expand this point in your Discussion*.

It is very likely that the observed homologous and heterologous blood-stage protection after CPS immunization with *P. chabaudi* AS infected mosquito bites (Figures 4 and 5) is due to transient blood-stage exposure during immunization, which persists for 3 to 4 days after each immunization (Figure 2). CPS immunized human volunteers experience a similar pattern of transient blood-stage parasitemia during the immunization procedure (49, 7, 8). Furthermore the small number of infected hepatocytes after *P. chabaudi* AS mosquito transmission (see comment #1) is unlikely to induce cross-protective responses directed against blood-stage parasites (given that they do not even induce stage-specific responses).

Low-level transient blood-stage exposure was shown previously in mice (16, 57, 1, 22) and humans (46) to induce long-lived blood-stage immunity. Here we show that a significant protective effect of CPS-induced blood-stage immunity is apparent after 6 to 7 erythrocytic replication cycles following challenge (Figures 4 and 5).

It is plausible that this would similarly be true in CPS immunized humans, but cannot be detected because of the need to treat as soon as thick blood film detectable blood parasitemia is reached (typically between the third and fourth erythrocytic replication cycle after challenge, [7]).

One volunteer, classified as protected from mosquito bite challenge infection by the absence of a positive slide and no clinical symptoms, had 457 parasites per ml as determined by retrospective qRT PCR at 21 days post challenge just prior to presumptive drug treatment (7). In addition to a profound reduction of liver-stage parasitemia and prolonged liver-stage development, blood-stage immunity may have contributed to this delay. Conversely this volunteer had over 1.3 million parasites in his blood stream (assuming 3l of blood), which may also mean that blood-stage immunity was not acquired.

Because of the difficulties to evaluate partial blood-stage protection in human volunteers, animal models like the here presented novel *P. chabaudi*-C57BL/6 mouse model are particularly valuable.

The Discussion relating to blood-stage immunity has been expanded.

*4) The CPS immunization protocol in mice protects against blood-stage infection (including heterologous strain) only when the challenge is delivered by mosquito bite (and not serial blood passage). This suggests immunity is raised against variant antigens that are shared between the heterologous strains following mosquito bite delivery. Although the data show protection against these genetically distinct strains of *P. chabaudi, *the potential for immune escape suggested by these results warrants further discussion*.

Figures 4 and 5 clearly show that CPS immunization with *P. chabaudi* AS infected mosquito bites can protect against mosquito bite challenge with the AS and the more virulent CB strain and importantly also against direct blood challenge with recently mosquito transmitted blood-stage parasites (Figure 4). Conversely CPS-induced blood-stage immunity was almost overridden after direct blood challenge with 26-32 times serially blood-passaged parasites (Figure 4). This suggests that protective immune responses target antigens expressed by mosquito transmitted blood-stage parasites but not by serially blood passaged parasites. Serially blood passaged parasites can hence escape from protective blood-stage immunity elicited by CPS immunization with mosquito bites.

One group of *Plasmodium* genes, whose expression is altered by mosquito transmission during blood-stage infection is the *Plasmodium* interspersed repeat gene family (*pir*); termed *cir* in *P. chabaudi* (30). While only very few *cir* genes are transcribed after serial blood-passage, transcription of more than half of all *cir* genes was increased after mosquito transmission (52). Furthermore *P. berghei pir* genes are transcribed not only in blood-stage parasites but also in ookinetes (44) and liver-stages (personal communication B.M. Franke-Fayard and C.J. Janse, Leiden University Medical Center, The Netherlands), making them potential candidates for the induction of cross-stage immune responses. An investigation into the shared PIR proteins expressed by liver- and blood-stage parasites may hence provide valuable information for multi-stage malaria vaccine development.

Since serially blood passaged parasites can i) escape the protective immune response, ii) would never be encountered in an endemic setting, iii) elicit different immune responses, iv) transcribe different genes and thus v) display enhanced virulence compared to mosquito transmitted blood-stage parasites (53), we believe that it is essential to always evaluate blood-stage protection in the context of mosquito transmission.

The Discussion has been extended to highlight the potential for immune escape and the role of the *Plasmodium* interspersed repeat gene family after challenge with serially blood passaged parasites.

*5) A careful editing of the manuscript was suggested to avoid a possible confusion*:

*a) In the Introduction, the authors state that CPS provides both heterologous and pre-erythrocytic immunity, but later state that CPS “abrogate” pre-erythrocytic immunity (in the end of the Results section)*.

Both references have been amended to avoid confusion.

*b) A disunity in the message is that low dose CPS immunization by mosquito bite offers no pre-erythrocytic protection but some homologous or heterologous erythrocytic immunity; high dose sporozoite challenge iv or no drug, “self-cured”mosquito bite or* ip *blood stage infection provide pre-erythrocytic immunity.*

We added Table 1 to clarify the effect of low and high antigen dose (the number of immunizing sporozoites and liver-stage parasites as well the amount of blood-stage parasites) for the acquired protection. What is perceived as a disunity of message reflects the complexity of immunity against the different life cycle stages of the malaria parasite (please see the Discussion). That immunity is not necessarily life cycle stage specific also needs to be taken into account.

Our findings are comprehensively discussed and summarized in Table 1.